# Thermal biology of two tropical lizards from the Ecuadorian Andes and their vulnerability to climate change

**Estefany S. Guerra-Correa**[1]*, **Andrés Merino-Viteri**[1,2], **María Belén Andrango**[1], **Omar Torres-Carvajal**[1]

1 Escuela de Ciencias Biológicas, Museo de Zoología, Pontificia Universidad Católica del Ecuador, Quito, Pichincha, Ecuador, 2 Escuela de Ciencias Biológicas Laboratorio de Ecofisiología, Pontificia Universidad Católica del Ecuador, Quito, Pichincha, Ecuador

* estefy92guerra@gmail.com

## Abstract

This study aims to analyze the thermal biology and climatic vulnerability of two closely related lizard species (*Stenocercus festae* and *S. guentheri*) inhabiting the Ecuadorian Andes at high altitudes. Four physiological parameters—body temperature ($T_b$), preferred temperature ($T_{pref}$), critical thermal maximum ($CT_{max}$), and critical thermal minimum ($CT_{min}$) —were evaluated to analyze the variation of thermophysiological traits among these populations that inhabit different environmental and altitudinal conditions. We also evaluate the availability of operative temperatures, warming tolerance, and thermal safety margin of each population to estimate their possible risks in the face of future raising temperatures. Similar to previous studies, our results suggest that some physiological traits ($CT_{max}$ and $T_b$) are influenced by environmental heterogeneity, which brings changes on the thermoregulatory behavior. Other parameters ($T_{pref}$ and $CT_{min}$), may be also influenced by phylogenetic constraints. Moreover, the fluctuating air temperature ($T_{air}$) as well as the operative temperatures ($T_e$) showed that these lizards exploit a variety of thermal microenvironments, which may facilitate behavioral thermoregulation. Warming tolerance and thermal safety margin analyses suggest that both species find thermal refugia and remain active without reducing their performance or undergoing thermal stress within their habitats. We suggest that studies on the thermal biology of tropical Andean lizards living at high altitudes are extremely important as these environments exhibit a unique diversity of microclimates, which consequently result on particular thermophysiological adaptations.

## Introduction

Vertebrate ectotherms, such as reptiles, are one of the most threatened groups due to climate change [1, 2] because their biology is intimately tied to temperature and they also exhibit rapid, and sensitive physiological responses towards environmental perturbations [3, 4, 5, 6]. Indeed, their performance on ecologically-relevant tasks are temperature-dependent,

**Data Availability Statement:** All operative temperatures from study sites and all individual thermophysiological data files are available on the

Open Science Framework database through link:
https://osf.io/yxgbs/quickfiles.

**Funding:** This research was funded by Pontificia
Universidad Católica del Ecuador (Grant number:
K13060). The funder had no role in study design,
data collection and analysis, decision to publish, or
preparation of the manuscript.

**Competing interests:** The authors have declared
that no competing interests exist.

increasing gradually from a critical thermal minimum ($CT_{min}$) to an optimal temperature ($T_{opt}$) and then falling rapidly as body temperature approaches the critical thermal maximum ($CT_{max}$) [3, 7, 8]. In spite of their thermally sensitive performance, ectotherms are not fully at the whim and mercy of their environments. Through behavioral thermoregulation, they can preferentially select microclimates within their habitats that match their preferred conditions [9, 10]. The effectiveness of this thermoregulatory behavior depends on the availability of suitable thermal microclimates; nonetheless, if these are scarce, this behavior can have effects on activity patterns, habitat selection, and spatial distribution [11, 12].

In addition to the constraints of physiology and behavior, there are features that make some lineages of ectotherms more sensitive to climate change than others. Compared to species living at higher latitudes, tropical species usually have more constricted physiological tolerance ranges and a reduced ability to thermoregulate behaviorally [1, 13]. For instance, some tropical and subtropical forest lizards are facing more challenges to withstand short-term temperature shifts because they live in habitats where operative temperatures are relatively less variable, having less options for thermoregulation as they have evolved narrower thermal tolerance in contrast to species of temperate zones [1, 14]. Whereas, considering open tropical habitats, where operative temperatures may reach a huge range of available body temperatures to thermoregulate, lizards can be overwhelmed by the extreme temperatures in their microhabitats [15].

At a global scale, tropical montane ecosystems are one of the most important areas of species richness and endemism, but also one of the biodiversity hotspots most vulnerable to global warming [16]. A particular case of study on southern South America indicate that the maximum temperature increase occurs between 4 and 5˚C in tropical and subtropical regions and the largest warming is generally found over the Andes on two seasons in the year [17]. Also, projections of climate change for the 21st century have shown significant warming in the tropical Andes, which is enhanced at higher elevations [18]. These radical changes on temperature can provoke negative consequences on lizards since behavioral adjustments will probably not be enough to avoid overheating and consequently life history aspects could be compromised [19].

The genus *Stenocercus* is one of the most geographically and ecologically widespread lizard taxa in South America [20], with 68 species occurring from northern Venezuela and Colombia to central Argentina, between sea level and 4000 m [21]. To our knowledge, this is the second study that evaluates the thermal biology of tropical high-Andean *Stenocercus* species. Thermophysiological studies on temperate Andean lizards, which presumably have analogous thermophysiological traits to tropical Andean lizards, have revealed interesting thermophysiological patterns. For example, thermoregulatory strategies of *Liolaemus* species may rely on both air and substrate temperature [22, 23], and also swapping between shaded areas and sites exposed to direct sunlight [24]. Moreover, it has been debated whether the body (field) temperature of this group of lizards is driven by a labile evolution in which the environmental thermal gradient causes a directional selection, or it responds to the phylogeny and thus thermal physiology is evolutionarily conservative [22, 23, 24, 25, 26, 27, 28].

In this study, we analyze the thermal physiology and vulnerability to climate change of three populations of two closely related lizard species living at high altitudes in the Ecuadorian Andes, *Stenocercus guentheri* and *S. festae*. Given that these populations inhabit different environmental and altitudinal conditions, we expect them to differ in their thermophysiological traits. The difference between these traits can allow us to identify which populations may be most threatened by environmental changes. To test this hypothesis, we evaluate four physiological parameters among all populations–body temperature ($T_b$), preferred temperature ($T_{pref}$), critical thermal maximum ($CT_{max}$), and critical thermal minimum ($CT_{min}$). We then

assess the availability of operative temperatures, warming tolerance, and thermal safety margin of each population to evaluate the possible risks these populations have in the face of global warming.

## Materials and methods

### Study species and sites

We studied two species of *Stenocercus* lizards, which are active thermoregulators [29, 30]. *Stenocercus guentheri* occurs in the northern Andes of Ecuador between 2135 and 3890 m, inhabits dry and humid premontane and montane forests, and is commonly found over rocks or nags (S1A Fig) [30, 31]. This species is under the Least Concern category of the IUCN [32]. For this species, we chose two sites at Pichincha Province that were different in elevation and climatic conditions. The first site was Jerusalem Recreational Park and Protected Forest (Jerusalem RP) (00˚ 00' 29.77" N, 78˚ 15' 41.25" W, 2278 m). This location is one of the last patches of dry Andean matorral and dry montane forest in the northern Andes (S2A Fig), where the average yearly temperature is about 16˚C, ranging from an average minimum of 10˚C to a maximum of 24˚C [33, 34]. The second site, Calacalí (00˚ 00' 26.5" S, 78˚ 31' 04.2" W, 2950 m), is characterized by crops, natural vegetation and small patches of cloud forest (S2B Fig) [35]. At this site, the annual mean air temperature is 13.6˚C, ranging from an average minimum of 7˚C to a maximum of 20˚C [34].

The second species, *Stenocercus festae*, assessed as Vulnerable due to the continuing decline of the quality of its habitat [36], occurs in the Andes of southern Ecuador, between 1050 and 3200 m (S1B Fig) [31]. It inhabits low dry montane, wet montane and humid sub-Andean forests and is frequently found at the base of small bushes or nags. The study site for this species was carried out at La Paz Scientific Station (La Paz), Azuay Province (03˚ 20' 18" S, 79˚ 10' 18.01" W, 3100 m), for which we obtained permission from Universidad del Azuay. Pine plantations, pastures, crops and areas of natural vegetation mainly cover this locality (S2C Fig) [37]. This site is characterized by fog and cold winds, where the annual mean air temperature is 11˚C, ranging from an average minimum of 6˚C to a maximum of 15˚C [34, 37].

### Specimen and field data collection

In 2015, we collected 26 adult specimens of *Stenocercus guentheri* from Calacalí between May and August, and 21 specimens of *S. festae* on February, May, and June. Data of 27 adult specimens of *S. guentheri* from Jerusalem RP were taken from Andrango et al. [38]. Specimens were collected under permit N˚ 003–15 IC-FAU-DNB/MA issued by Ministerio del Ambiente del Ecuador.

Sampling was carried out from 9:00 to 16:00 using a Cabela's Panfish Pole with waxed dental floss moored for noosing. All individuals were collected while active outside their retreats in their habitats. Body temperature ($T_b$) was measured externally over the cloacal region of the lizard with a T-type thermocouple connected to an Omega HH603A digital thermometer. To avoid heat transfer to the animals, $T_b$ was taken within 20 s of capture by handling individuals by the head while still noosed.

### Thermophysiological data

Before thermophysiological experiments, lizards were placed individually for a maximum of two days at room temperature after capture in terraria (0.28 m long x 0.175 m wide x 0.17 m high), with water ad libitum. Thermophysiological data of *S. festae* and *S. guentheri* from Calacalí were recorded in the field and in the laboratory, respectively. We started with the least

invasive experiments ($T_{pref}$), followed by $CT_{min}$ and $CT_{max}$, with a 24-hour interval between each experiment to reduce animal suffering and stress. Once thermophysiological data was measured, specimens were sacrificed following standard protocols [39], reviewed and approved by Ministerio del Ambiente del Ecuador before issuing the collecting permit. All individuals were deposited in the collections of the Museo de Zoología from Pontificia Universidad Católica (QCAZ), Quito, Ecuador. None of the specimens died while we performed thermophysiological experiments. Approval by an Animal Ethics Committee for experimental manipulations is not required by QCAZ. However, this study was evaluated and approved by the DGA (Dirección General Académica) of the Pontificia Universidad Católica del Ecuador in accordance with the guidelines for environmental and social impacts of research projects. The DGA committee evaluated this project (K13060) to determine observance of its norms for ethical scientific research.

To determine $T_{pref}$ we built an eight-lane wooden track of 1 m long x 0.12 m wide x 0.2 m high [12]. In each lane, we installed a 100-watt light bulb at one end to create a thermal gradient, along which the lizard was free to move. The thermal gradient obtained was from 40 to 23°C for *S. guentheri* from Calacalí and from 35 to 12°C for *S. festae*. Body temperatures were recorded in real time by the program Omega Logging Recorder every 30 seconds for 2 hours through a T-type thermocouple placed anteriorly to the cloacal area and secured with Micropore surgery tape. In each trial, we measured the $T_{pref}$ of 2 to 3 individuals and a maximum of 8 individuals per day, starting from 9:00. Every time we began with a new experiment, lizards were released in the middle of the lane with a 15-minute period of acclimation.

$CT_{min}$ and $CT_{max}$ were independently measured by placing each lizard in a clear glass chamber immersed in water cooled and heated at an approximate rate of 1.0–1.5°C per minute. A T-type thermocouple attached anteriorly to the cloacal area and secure with Micropore surgery tape, recorded the temperature at which the individual was unable to right itself when flipped onto its back [38, 40].

Because our data did not meet the assumptions of normality and variance-homogeneity, we analyze the variation of the thermophysiological data among the three study populations using a Kruskal-Wallis Test. These statistical analyses were performed in SPSS (IBM Corporation, Version 19.0).

## Operative temperature and air temperature data

To estimate the operative temperature ($T_e$), defined by Bakken [41] as a thermal index that represents the set of body temperatures a lizard experiences at it's spatial scale, we used physical models that simulate a lizard's phenotype in size and color [42, 43, 44]. Each model consisted of gray segment of PVC pipe, 10 cm long and 2 cm in diameter, connected to a two-channel HOBO Pro v2 U23-003 data logger that recorded temperature data every 5 minutes. The external temperature probes of all data loggers were inserted into the PVC models, sealed with liquid silicone and placed onto the soil at different microhabitats used by the species. One model was placed over the ground at shaded refuge sites (e.g. under small bushes, rocks or logs) and the other model registered open sites exposed to direct sunlight, as bare rocks and logs where active lizards were observed (S2C Fig). To determine if these models represent the body temperature of a lizard, we registered simultaneously temperature data of the models and body temperature of lizards in the field. Then, we conducted a two-sample t-test that showed there were no significant differences between the mean body temperature of lizards (mean = 29.6°C, SD = 0.74) and the mean temperature of the models (mean = 30.1°C, SD = 1.37); N = 27, t = -1.695, $P$ = 0.097. Physical models were placed on sites representing four randomly chosen microhabitats used by *S. guentheri* from Calacalí and *S. festae*. Data

loggers were launched for five months in Calacalí (between 15 May and 22 September 2015) and for nine months in La Paz (between 6 October 2014 and 4 June 2015). Andrango et al. [38] registered the $T_e$ of *S. guentheri* from Jerusalem RP for 5 months, between 23 March and 23 August 2014, at five representative microhabitats used by the species. The operative temperature-sampling periods differ because they correspond to the hottest months of each population [34]. The mean operative temperature per hour for each site was obtained by independently averaging the temperature values of shelters and sunlight exposed sites (i.e., minimum and maximum operative temperatures, respectively) and, subsequently, averaging those two values.

An additional data logger was placed under the shadow at approximately 1.50 meters above the ground to record air temperature ($T_{air}$). Owing to logistics, we obtained $T_{air}$ data for 3 months in Calacalí (between 5 May and 17 July 2015) and Jerusalem RP (between 28 September and 28 November 2017), and 10 months in La Paz (between 6 October 2014 and 22 October 2015, excluding January and September 2015). To avoid any biases due to the difference of sampling time at each site, we extracted the air temperature from the climatological data provided by Climate-Data.org database (weather data collected between 1982 and 2012) [34], which show similar $T_{air}$ to the $T_{air}$ data registered in the field (Fig 1).

### Assessment of vulnerability to climate change

We analyzed the availability of operative temperatures each population can exploit by developing a scripting routine in R software (R Core Team, Version 3.3.2). First, we calculated the mean maximum operative temperature per hour for models exposed to sunlight (Max $T_e$) and the mean minimum operative temperature per hour for models in shaded sites (Min $T_e$). Then, we plotted values of body temperature ($T_b$) with the corresponding hour of capture, as well as the critical thermals ($CT_{max}$ and $CT_{min}$) and $T_{pref}$.

To assess the vulnerability to climate warming of these lizards, we calculated the warming tolerance (WT) and the thermal safety margin (TSM) of each population. WT quantifies the average amount of environmental warming an ectotherm may tolerate before its performance drops to fatal levels [3]. To obtain WT values, we calculated the difference between the minimum $CT_{max}$ value of all the assessed individuals and the mean operative temperature of models placed in shaded areas at the hottest hours of the day (12:00 *S. guentheri* from Jerusalem, 11:00 *S. guentheri* from Calacalí and 13:00 for *S. festae*) [$T_e$s; 3, 14]. TSM is the difference between a lizard's optimal temperature ($T_{opt}$) and the mean hourly minimal $T_e$ in the shade [$T_e$s min; 3]. Here we used $T_{pref}$ instead of $T_{opt}$, since $T_{pref}$ most often sub-estimates $T_{opt}$ and thus gives more suitable interpretations on the effects of climate change on lizards' fitness [45]. For the latter analyses, we used the minimum values of $T_e$ of models exposed to shade sites because they are better descriptors of the thermal microenvironment these lizards have in order to avoid warming [15].

## Results

### Thermophysiological parameters

*Stenocercus guentheri* from Calacalí had the highest body temperature (32.3˚C) of the three studied populations and was significantly different from *S. festae* ($P = 0.004$), which had the lowest temperature (28.2˚C). However, *S. guentheri* from Jerusalem RP showed the highest mean $T_{pref}$ (31.8˚C) and was significantly different from both *S. festae* and *S. guentheri* from Calacalí ($P < 0.01$). Regarding critical thermal limits, *S. festae* showed the highest mean $CT_{max}$ (46.8˚C) and was significantly different from both populations of *S. guentheri* ($P < 0.01$); whereas *S. guentheri* from Calacalí and *S. festae* showed the lowest $CT_{min}$ values and were

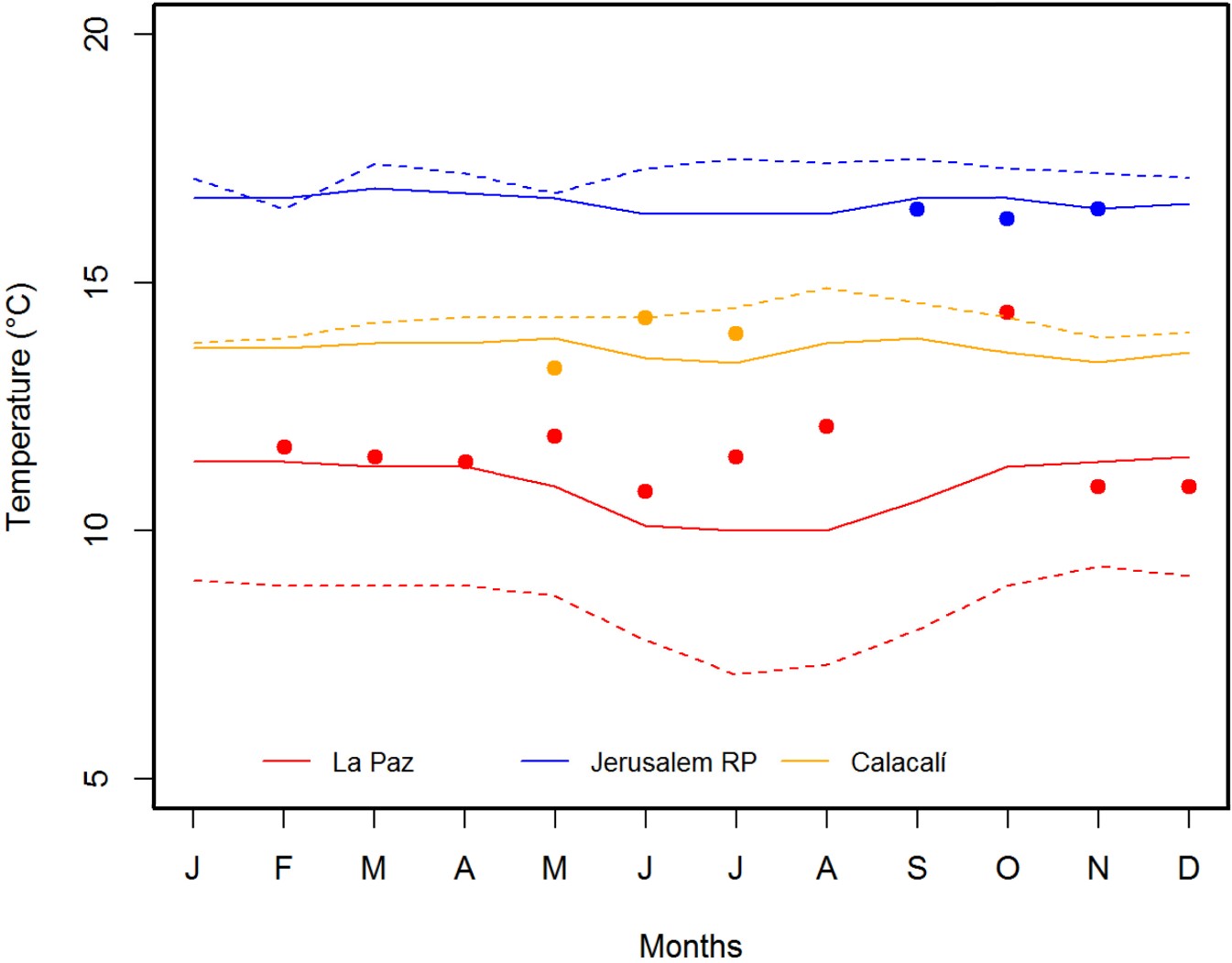

**Fig 1. Air temperature of the three studied sites from three different sources.** The solid lines show the mean air data extracted from Climate-data.org website while the dotted lines show the mean air data provided by INAMHI from closely meteorological stations to the study sites. The colored dots represent the mean air temperature registered with the data logger placed in the field at each study site.

significantly different from *S. guentheri* from Jerusalem RP ($P = 0.01$; $P < 0.01$, respectively) (Tables 1 and 2).

### Environmental profile

The mean $T_e$ recorded at sun exposed sites (max $T_e$) and shadow sites (min $T_e$) showed higher values for *S. guentheri* at Jerusalem RP (max $T_e = 23.4°C$, min $T_e = 17.7°C$) than populations of *S. guentheri* at Calacalí (max $T_e = 18.1°C$, min $T_e = 15.8°C$) and *S. festae* (max $T_e = 14.6°C$, min $T_e = 11.9°C$). The mean $T_e$ was $20.9°C$ for *S. guentheri* at Jerusalem RP, $16.9°C$ for *S. guentheri* at Calacalí and $13.2°C$ for *S. festae*.

The air temperature provided by Climate-Data.org database [34] showed that La Paz Scientific Station, in Azuay province, is much colder (min = 5.8°C, max = 16.6°C, mean = 10.9°C)

**Table 1. Thermophysiological data for adults of the three *Stenocercus* populations analyzed this study.**

| Population | $T_b$ | $T_{air}$ | $T_{sub}$ | $T_{pref}$ | $CT_{max}$ | $CT_{min}$ | Thermal breadth |
|---|---|---|---|---|---|---|---|
| *S. guentheri* | ($n = 17$) | | | ($n = 27$) | ($n = 25$) | ($n = 23$) | |
| Jerusalem RP | 31.5 ± 2 .5 | 27.4 ± 3.3 | 32.3 ± 4.7 | 31.8 ± 2.0 | 43.8 ± 2.4 | 7.9 ± 1.9 | 35.9 |
| | 24.3–34.7 | | | 27.0–35.1 | 37.0–47.0 | 5.4–11.6 | |
| *S. guentheri* | ($n = 26$) | | | ($n = 26$) | ($n = 26$) | ($n = 26$) | |
| Calacalí | 32.3 ± 2.5 | 24.3 ± 3.8 | 32.8 ± 8.7 | 22.6 ± 0.9 | 44.4 ± 2.2 | 6.5 ± 1.4 | 37.9 |
| | 25.1–35.6 | | | 19.8–24.2 | 39.5–47.6 | 4.2–11.1 | |
| *S. festae* | ($n = 20$) | | | ($n = 21$) | ($n = 21$) | ($n = 21$) | |
| La Paz | 28.2 ± 5.5 | 17.6 ± 2.4 | 26.9 ± 5.3 | 22.7 ± 5.0 | 46.8 ± 1.5 | 5.6 ± 2.2 | 41.2 |
| | 16.0–34.8 | | | 14.5–31.9 | 43.4–49.2 | 2.6–11.2 | |

Mean body temperature ($T_b$), mean air temperature ($T_{air}$) and substrate temperature ($T_{sub}$) at the moment of capture, mean preferred temperature ($T_{pref}$), mean critical thermal maxima ($CT_{max}$) and minima ($CT_{min}$), and thermal breath ($CT_{max}$–$CT_{min}$). Temperature is in ˚C, sample size (first line), mean ± SD (second line), and temperature ranges are presented for each *Stenocercus* population.

than Jerusalem RP (min = 8.6˚C, max = 24.4˚C, mean = 16.6˚C) and Calacalí (min = 6.5˚C, max = 20.9˚C, mean = 13.7˚C) (Fig 1).

Daily operative temperatures were higher around midday in all three study populations and sunlight-exposed areas reached temperatures higher than lizards' $CT_{max}$ values. Shadowed areas at La Paz were cooler than the average $CT_{min}$ of *Stenocercus festae* all day. In contrast, min $T_e$ values were cooler than the average $CT_{min}$ only early in the morning at Jerusalem RP and not a single time on the day in Calacalí. All three sites have microhabitats with operative temperatures lower than the $T_{pref}$ of the species during the whole day (Fig 2).

### Warming tolerance and thermal safety margin

Warming tolerance was higher for *Stenocercus festae* (WT = 43.4˚C– 3.5˚C = 39.9˚C) than both populations of *S. guentheri*—WT = 37.0˚C– 17.8˚C = 19.2˚C at Jerusalem RP and WT = 39.5˚C– 12.2˚C = 27.3˚C at Calacalí.

Thermal safety margins showed positive values for the three populations analyzed. TSM = 31.8˚C– 5.6˚C = 26.2˚C for *S. guentheri* from Jerusalem RP, TSM = 22.7˚C– 4.5˚C = 18.2˚C for *S. guentheri* from Calacalí, TSM = 22.7˚C– 1.2˚C = 21.5˚C for *S. festae* from La Paz.

## Discussion

Phylogenetic and environmental adaptive forces seem to be responsible for shaping the thermal biology of other lizard taxa, such as *Liolaemus* [24], a lineage suggested to exhibit adaptive

**Table 2. Kruskal-Wallis test results for thermophysiological data of three *Stenocercus* populations analyzed in this study.**

| Populations | | $T_b$ | $T_{pref}$ | $CT_{max}$ | $CT_{min}$ |
|---|---|---|---|---|---|
| *S. guentheri* (Jerusalem RP) | $\chi^2$ | 3.758 | 29.762*** | 19.180*** | 12.507*** |
| vs. *S. festae* | $P$ | 0.053 | <0.001 | <0.001 | <0.001 |
| *S. guentheri* (Jerusalem RP) | $\chi^2$ | 1.575 | 39.022*** | 1.322 | 6.681** |
| vs. *S. guentheri* (Calacalí) | $P$ | 0.209 | <0.001 | 0.250 | 0.010 |
| *S. guentheri* (Calacalí) | $\chi^2$ | 8.226** | 0.148 | 13.869*** | 5.049 |
| vs. *S. festae* | $P$ | 0.004 | 0.700 | <0.001 | 0.025 |

Variables include mean body temperature ($T_b$), mean preferred temperature ($T_{pref}$), mean critical thermal maxima ($CT_{max}$), and minima ($CT_{min}$). Significant differences are defined as *** if $P < 0.001$ and ** if $P < 0.01$.

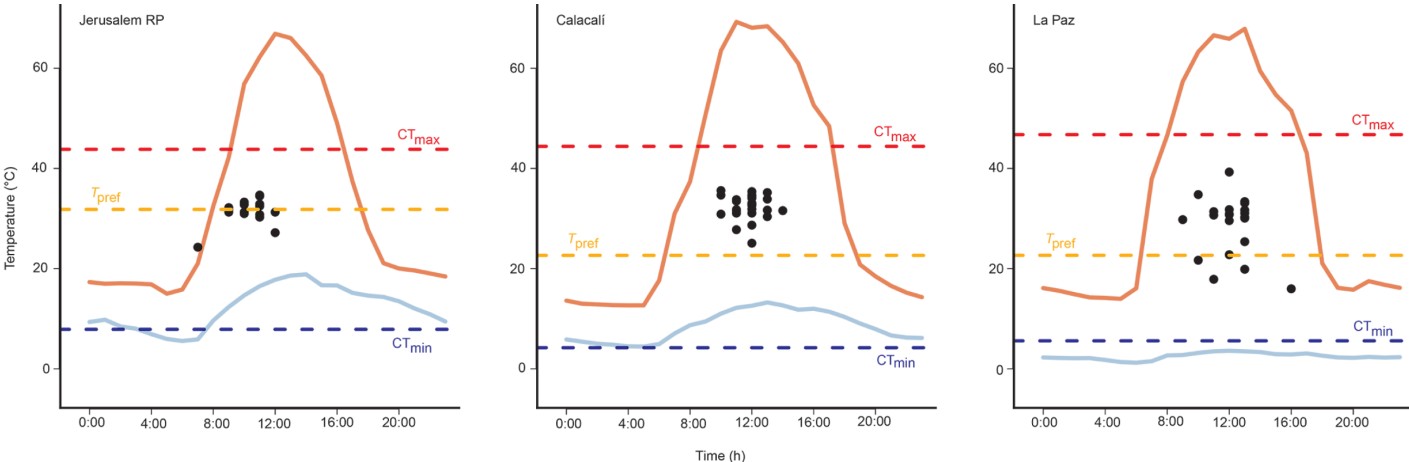

**Fig 2. Range of available operative temperatures in the three study sites.** The solid lines show the fluctuation per hour of the mean maximum and mean minimum operative temperatures registered in the field. Black dots represent the $T_b$ of each individual plotted against the corresponding hour of capture, while the dashed lines show $T_{pref}$, $CT_{min}$ and $CT_{max}$ mean values for each population.

radiation parallel to *Stenocercus* on different areas of South America (i.e. similar phenotypic variation patterns) [46]. As we hypothesize, there are significant inter- and intraspecific differences in some thermophysiological traits between the three populations studied herein. *Stenocercus festae* and *S. guentheri* from Calacalí differed significantly in $T_b$ values, whereas *S. guentheri* from Jerusalem RP showed significant differences on $T_{pref}$ values with both *S. festae* and *S. guentheri* from Calacalí. $CT_{max}$ was significantly different between *S. festae* and both populations of *S. guentheri*, whereas *S. guentheri* from Calacalí and *S. festae* were significantly different from *S. guentheri* from JPR in $CT_{min}$. This suggests that some differences in physiological traits among populations/species of *Stenocercus* are caused by climatic factors that change systematically along altitude (La Paz 3100 m, Calacalí 2950 m, and Jerusalem RP 2278 m), creating selective pressures on physiology [47, 48]. Additionally, we found that our study sites had operative temperatures that could provide complex microclimatic mosaics, which is congruent with the idea that tropical landscapes present enough thermal microhabitats for species to evolve heat tolerances through behavioral specialization [49]. Nevertheless, studies on the thermal physiology of other species of *Stenocercus* are necessary to better understand the main evolutionary forces acting on the thermal biology of this large radiation of Andean lizards.

Two populations in this study show higher $T_b$ than $T_{pref}$ (Table 1). On one hand, high $T_b$ values are explained by the hours of capture of the individuals, which for most of the lizards assessed in this study correspond to hottest time of the day. Besides, Pearson [50] has shown that tropical lizards living at high altitude can achieve high body temperatures by using solar radiation. On the other hand, $T_{pref}$ experiments were performed along the day from 9:00, covering hours at which lizards were not found basking to direct sunlight in the field. Thus, it is important to consider that these lizards are probably used to performing activities at low temperatures given that the mean air temperature is 13.7°C and 10.9°C for Calacalí and La Paz, respectively. Additionally, we must consider the fact that this field of research has progressed tremendously over the last couple of decades [e.g. 3, 7, 8, 9, 15, 41, 42], so there may be some limitations on measuring the thermal preferences of these lizards [51].

Gunderson and Leal [52] suggested that activity periods are more sensitive to temperature than the whole-organism physiological traits; as a consequence, behavioral traits can shape the

way a population exploits the thermal heterogeneity of its habitat. Our data on hourly maximum and minimum $T_e$ suggest that the populations of *Stenocercus* included in this study have a great availability of thermal microenvironments to exploit, and thus no evident thermal restriction to perform their daily activities (Fig 2). Still, we recognize that more behavioral studies are needed to better understand how lizards are using thermal microenvironments.

Our warming tolerance analyses suggest that none of the studied populations will face a thermal deficit (temperatures over their $CT_{max}$). Furthermore, thermal safety margins analyses showed positive values for all the populations implying that they can find thermal refugia and may remain active without reducing their performance or undergoing thermal stress [14].

Species from warm environments may be under a higher risk of extinction than species living in cooler sites, as warmer environments may increase maintenance energy costs while simultaneously constraining activity time [53]. Considering that La Paz is the coldest site in this study (Fig 1) and that the population of *S. festae* inhabiting La Paz has a thermal breadth of 41.2°C (35.9°C *S. guentheri* from Jerusalem RP and 37.9°C *S. guentheri* from Calacalí), we suggest that the two populations of *S. guentheri* studied herein will be more sensitive to global warming than *S. festae* because they have a narrower thermal tolerance breadth and as proposed by Grigg and Buckley [54] they are exposed to temperatures closer to their physiological limit.

Although there are good studies about the thermophysiology of Andean lizards living at high altitudes [e.g. 11, 24, 43, 55], knowledge regarding the thermal biology of tropical high-Andean species is still scant. These lizards are of special concern because, though warming in the tropics is relatively small in magnitude compared with higher latitudes [3], tropical ectotherms have been suggested to be living very close to their optimal temperature, which increases the risk that environmental temperature changes affect them. Furthermore, tropical high Andes are already experiencing significant shifts in temperature, rainfall regimes and seasonal weather patterns [56, 57] that may have negative effects on its biodiversity. In this context, we consider that it is important to gather additional data, such as thermal acclimation, adaptation, dispersion capacities and behavioral modifications, that could lead us to a better understanding of the dynamics involved in the responses of Andean ectotherms to climate change.

## Supporting information

**S1 Fig. Photos of the studied species.** (A) *Stenocercus guentheri* (from left to right): QCAZR 13862 (female), QCAZR 13864 (male) (B) *Stenocercus festae* (from left to right): QCAZR 14014 (female), QCAZR 14022 (male).
(TIF)

**S2 Fig. Representative pictures of the three studied sites.** (A) Jerusalem Recreational Park and Protected Forest (Jerusalem RP), (B) Calacalí, (C) La Paz Scientific Station (La Paz) showing a physical model that record operative temperatures.
(TIF)

## Acknowledgments

We thank Priscila Barragán for her help in collecting thermal physiology and environmental data of *S. guentheri* from Calacalí, as well as all the volunteers who assisted during fieldwork and data collection, especially Clemencia Correa, María José Navarrete, Andrés Mármol, Luis S. Ruiz, and Keyko Cruz. We also thank Boris Tinoco and Verónica Urgilés from Universidad del Azuay for facilitating fieldwork at La Paz Scientific Station. Special thanks to Martha

Muñoz, Pol Pintanel, and Felipe Vanegas for their helpful comments on earlier versions of this manuscript.

## Author Contributions

**Data curation:** Estefany S. Guerra-Correa.

**Formal analysis:** Estefany S. Guerra-Correa, Andrés Merino-Viteri, María Belén Andrango.

**Funding acquisition:** Omar Torres-Carvajal.

**Investigation:** Estefany S. Guerra-Correa, María Belén Andrango.

**Project administration:** Omar Torres-Carvajal.

**Supervision:** Andrés Merino-Viteri, Omar Torres-Carvajal.

**Writing – original draft:** Estefany S. Guerra-Correa.

**Writing – review & editing:** Estefany S. Guerra-Correa.

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
