## [Editor Report · Decision Letter 0]

25 Nov 2019

PONE-D-19-27706

Thermal biology of two tropical lizards from the Ecuadorian Andes and their vulnerability to climate change.

PLOS ONE

Dear Ms Guerra-Correa,

Thank you for submitting your manuscript to PLOS ONE. After careful consideration, we feel that it has merit but does not fully meet PLOS ONE’s publication criteria as it currently stands. Therefore, we invite you to submit a revised version of the manuscript that addresses the points raised during the review process.

I was not involved in the previous review of your manuscript, but thoroughly assessed the reviews during the first and second round of reviews, and assessed your response to reviewers in this version of the manuscript. I am satisfied that you appropriately responded to those comments. I did thoroughly review your manuscript myself, and have a list of minor comments that must be addressed, as well as two major points. See my review below. I look forward to seeing your revised manuscript.

We would appreciate receiving your revised manuscript by Jan 09 2020 11:59PM. To enhance the reproducibility of your results, we recommend that if applicable you deposit your laboratory protocols in protocols.io, where a protocol can be assigned its own identifier (DOI) such that it can be cited independently in the future. For instructions see: http://journals.plos.org/plosone/s/submission-guidelines#loc-laboratory-protocols

We look forward to receiving your revised manuscript.

Kind regards,

William David Halliday, Ph.D.

Academic Editor

PLOS ONE

Journal Requirements:

Additional Editor Comments:

These revisions seem to be fairly thoroughly done, and the English language has also been improved throughout. I found some minor issues, mostly regarding language, that should be fixed (see below).

One important point, in my mind, is that you compare two populations of one species and one population of the other species, but the interspecific comparison does not happen at the same site. Without this overlap, or at least without a much greater sample size of populations for each species, this cannot be treated as a comparative analysis between these two species and different sites, which is how I interpreted it while I read it. Rather, this should be framed as an analysis of how thermal physiology related to environmental parameters and risk of increased heat stress caused by global warming in three separate populations, one of which happens to be a different but related species. This must be clarified throughout (especially the last paragraph of the introduction and throughout the discussion).

Another crucial point is your warming analysis, which you appear to compare to climate change scenarios based on average annual temperature. But how can you truly assess how often Te would get at or above CTmax based on these coarse analyses? Temperature modeled at the coarse scale is not a great predictor of micro-habitat temperatures. At a minimum, you should be comparing the daily maximum temperatures from the climate change report, but I’m still not convinced of the validity of this comparison. If you’re going to make this comparison, you must be able to convince your readers of the argument.

Line 73: “overwhelm” should be “overwhelmed”

Line 74: “on” should be “in”

Line 77: Remove “As a matter of fact”.

Line 78: I assume this is average yearly temperature? Please clarify in the text. Also, the average change on its own won’t be what’s important, but rather the extreme temperatures. Make this point clear as well.

Line 93: “evolutionary conservative” should be “evolutionarily conservative”

Line 215: switch “incorporate itself” to either “right itself” or “flip itself”

Line 253-254: should read “Daily operative temperatures were higher around midday in all three study populations and sunlight-exposed areas reached temperatures higher than lizards' CTmax values.”

Line 256: remove “during”.

Also line 256: “In contrast, this was observed ...” Explicitly explain state what “this” is – this is a new sentence, so you cannot assume connectivity with the previous sentence. I assume you’re referring to Te being below CTmin only during the early morning.

General structure of Methods and Results. Explain the Thermophysiological parameters section before the section on Te, because in the section on Te, you specifically rely on the reader to already know something about the thermophysiological parameters (comparing Te to Tpref, for example).

Table 2 – strange formatting issue when converted to PDF. Please fix.

Line 332: “Thus, it” should be “Thus, it is”

Line 333: change “consider that probably these lizards are more used to perform activities” to “consider that these lizards are probably used to performing activities”

Line 336: “over the last years” – over the last how many years? Provide references directly after this statement to show how it has progressed. I understand this is in direct response to a comment from the previous review, but it is currently unsubstantiated.

Line 336: delete “or restrictions”

Line 348: Is this rise in 4.8 C the annual average? If so, this has nothing to do with daily thermal maximums, per se.

Line 362: improper use of semi-colon. Should be a comma. This isn’t the first place I found this issue in the manuscript, either, so please check thoroughly throughout.

Line 369: should read “we consider that it is important”
---

## [Author Response · Author response to Decision Letter 0]

31 Dec 2019

1. One important point, in my mind, is that you compare two populations of one species and one population of the other species, but the interspecific comparison does not happen at the same site. Without this overlap, or at least without a much greater sample size of populations for each species, this cannot be treated as a comparative analysis between these two species and different sites, which is how I interpreted it while I read it. Rather, this should be framed as an analysis of how thermal physiology related to environmental parameters and risk of increased heat stress caused by global warming in three separate populations, one of which happens to be a different but related species. This must be clarified throughout (especially the last paragraph of the introduction and throughout the discussion).

- We don’t intend the reader to interpret this study as a comparative analysis between these three populations. Indeed, as you stated, we present a study showing the thermal physiology of S. guentheri and S. festae. We clarify this point as you suggested with more emphasis in the introduction and the discussion. 

2. Another crucial point is your warming analysis, which you appear to compare to climate change scenarios based on average annual temperature. But how can you truly assess how often Te would get at or above CTmax based on these coarse analyses? Temperature modeled at the coarse scale is not a great predictor of micro-habitat temperatures. At a minimum, you should be comparing the daily maximum temperatures from the climate change report, but I’m still not convinced of the validity of this comparison. If you’re going to make this comparison, you must be able to convince your readers of the argument.

- Our warming analysis doesn’t compare scenarios based on average annual temperature. On our warming tolerance (WT) analysis “we calculated the difference between the minimum CTmax value of all the assessed individuals and the mean operative temperature of models placed in shaded areas at the hottest hours of the day (P 10: L 236-238)”. We are aware that for this analysis of warming tolerance a coarse scale is a not a great predictor of micro-habitat temperatures that’s why we used the operative temperature (Te) measured by physical models placed on microhabitats used by each population. 

3. Line 73: “overwhelm” should be “overwhelmed”

Corrected. 

4. Line 74: “on” should be “in”

Corrected.

5. Line 77: Remove “As a matter of fact”.

Corrected. 

6. Line 78: I assume this is average yearly temperature? Please clarify in the text. Also, the average change on its own won’t be what’s important, but rather the extreme temperatures. Make this point clear as well.

- We changed this part with more accurate references as follows: 

A particular case of study on southern South America indicate that the maximum temperature increase occurs between 4 and 5 °C in tropical and subtropical regions and the largest warming is generally found over the Andes on two seasons in the year [17]. Also, projections of climate change for the 21st century have shown significant warming in the tropical Andes, which is enhanced at higher elevations [18]. (P 4: L 77-81)

7. Line 93: “evolutionary conservative” should be “evolutionarily conservative”

Corrected. 

8. Line 215: switch “incorporate itself” to either “right itself” or “flip itself”

Corrected. 

9. Line 253-254: should read “Daily operative temperatures were higher around midday in all three study populations and sunlight-exposed areas reached temperatures higher than lizards' CTmax values.”

Corrected.

10. Line 256: remove “during”.

Corrected. 

11. Also line 256: “In contrast, this was observed ...” Explicitly explain state what “this” is – this is a new sentence, so you cannot assume connectivity with the previous sentence. I assume you’re referring to Te being below CTmin only during the early morning.

- We explain this part as follows: In contrast, min Te values were cooler than the average CTmin only early in the morning at Jerusalem RP and not a single time on the day in Calacalí. (P 13: L 295-297)

12. General structure of Methods and Results. Explain the Thermophysiological parameters section before the section on Te, because in the section on Te, you specifically rely on the reader to already know something about the thermophysiological parameters (comparing Te to Tpref, for example).

- Methods and Results have been modified as you suggested.

13. Table 2 – strange formatting issue when converted to PDF. Please fix.

Done.

14. Line 332: “Thus, it” should be “Thus, it is”

Changed. 

15. Line 333: change “consider that probably these lizards are more used to perform activities” to “consider that these lizards are probably used to performing activities”

Changed.

16. Line 336: “over the last years” – over the last how many years? Provide references directly after this statement to show how it has progressed. I understand this is in direct response to a comment from the previous review, but it is currently unsubstantiated.

- We modified this sentence and provided references that show how this area of research has progressed. 

Additionally, we must consider the fact that this field of research has progressed tremendously over the last couple of decades [e.g. 3, 7, 8, 9, 15, 38, 41], so there may be some limitations on measuring the thermal preferences of these lizards [51]. (P 15: L 343-345)

17. Line 336: delete “or restrictions”

Changed. 

18. Line 348: Is this rise in 4.8 C the annual average? If so, this has nothing to do with daily thermal maximums, per se.

- We deleted this part because, as you pointed out in this comment, this value is an annual average, and cannot give an accurate reference to the daily thermal maximum. 

19. Line 362: improper use of semi-colon. Should be a comma. This isn’t the first place I found this issue in the manuscript, either, so please check thoroughly throughout.

- We checked the use of semi-colon in the manuscript.

20. Line 369: should read “we consider that it is important” 

Corrected.

---

## [Editor Report · Decision Letter 1]

7 Jan 2020

Thermal biology of two tropical lizards from the Ecuadorian Andes and their vulnerability to climate change.

PONE-D-19-27706R1

Dear Dr. Guerra-Correa,

We are pleased to inform you that your manuscript has been judged scientifically suitable for publication and will be formally accepted for publication once it complies with all outstanding technical requirements.

With kind regards,

William David Halliday, Ph.D.

Academic Editor

PLOS ONE
---

## [Editor Report · Acceptance letter]

17 Jan 2020

PONE-D-19-27706R1 

Thermal biology of two tropical lizards from the Ecuadorian Andes and their vulnerability to climate change. 

Dear Dr. Guerra-Correa:

I am pleased to inform you that your manuscript has been deemed suitable for publication in PLOS ONE. Congratulations! Your manuscript is now with our production department. 

With kind regards,

on behalf of

Dr. William David Halliday 

Academic Editor

PLOS ONE